# Non-Apoptotic Caspase-3 Activation Mediates Early Synaptic Dysfunction of Indirect Pathway Neurons in the Parkinsonian Striatum

**DOI:** 10.3390/ijms23105470

**Published:** 2022-05-13

**Authors:** Tim Fieblinger, Chang Li, Elena Espa, M. Angela Cenci

**Affiliations:** 1Basal Ganglia Pathophysiology Unit, Department of Experimental Medical Science, Lund University, 223 62 Lund, Sweden; chang.li@med.lu.se (C.L.); elena.espa@med.lu.se (E.E.); 2University Medical Center Hamburg-Eppendorf, Institute for Synaptic Physiology, 20251 Hamburg, Germany

**Keywords:** Parkinson’s disease, striatum, caspase-3, spiny projection neurons, dendritic spines, long-term depression, Q-VD-OPh, mice

## Abstract

Non-apoptotic caspase-3 activation is critically involved in dendritic spine loss and synaptic dysfunction in Alzheimer’s disease. It is, however, not known whether caspase-3 plays similar roles in other pathologies. Using a mouse model of clinically manifest Parkinson’s disease, we provide the first evidence that caspase-3 is transiently activated in the striatum shortly after the degeneration of nigrostriatal dopaminergic projections. This caspase-3 activation concurs with a rapid loss of dendritic spines and deficits in synaptic long-term depression (LTD) in striatal projection neurons forming the indirect pathway. Interestingly, systemic treatment with a caspase inhibitor prevents both the spine pruning and the deficit of indirect pathway LTD without interfering with the ongoing dopaminergic degeneration. Taken together, our data identify transient and non-apoptotic caspase activation as a critical event in the early plastic changes of indirect pathway neurons following dopamine denervation.

## 1. Introduction

Caspases are a family of cysteine proteases classically associated with apoptosis (i.e., programmed cell death) [1,2,3]. However, accumulating evidence shows that caspases also fulfill non-apoptotic roles in both neurons and glial cells [4,5]. Studies in hippocampal neurons have shown that caspase-3 can mediate the pruning of axons, dendrites and spines in the absence of cell death [6,7,8]. Moreover, restricted activation of caspase-3 in dendrites has been linked to long-term depression (LTD) of Schaffer collateral-CA1 synapses [8]. In microglia and astrocytes, caspase activation regulates the transition to proinflammatory or reactive phenotypes [9,10,11], and cytosolic expression of active caspase-3 has been observed in various brain cells even under basal conditions [5,12]. In models of Alzheimer’s disease, non-apoptotic caspase-3 activation has been shown to mediate hippocampal synaptic dysfunction, including altered AMPA receptor phosphorylation, loss of dendritic spines, and LTD deficits [6,13].

Loss of dendritic spines and deficits in LTD induction also occurs in striatal neurons in animal models of Parkinson’s disease (PD), a neurodegenerative disorder with typical motor symptoms caused by the degeneration of nigrostriatal dopaminergic projections [14,15,16,17,18,19,20,21]. Using toxin-based models of PD, previous studies have reported high levels of caspase-3 activity in the striatum [22,23]. In contrast, a non-symptomatic genetic model of autosomal recessive PD (the *PINK1* knockout mouse) shows normal striatal levels of caspase-3 activity at baseline, but an activity-dependent reduction of caspase-3 activity appears to be causally linked to synaptic plasticity deficits in this model [24]. These studies warrant further investigations into the functional role of caspase-3 in the pathophysiology of PD.

In the present study, we set out to investigate whether caspase-3 plays a role in the loss of dendritic spines and the concomitant synaptic deficits that develop in striatal spiny projection neurons (SPNs) upon the loss of dopaminergic innervation. Loss of SPN spines and glutamatergic synapses have been well-documented in all animal models of PD exhibiting severe nigrostriatal dopamine (DA) depletion [15,16,19,25,26], and have moreover been detected post-mortem in PD patients (reviewed in [27]).

Striatal SPNs divide into two major groups, called direct and indirect pathway SPNs (dSPNs and iSPNs, respectively). These two populations differ in their respective projection targets, expression of DA receptors, physiological properties, and response to DA denervation [27,28,29]. In models of PD, spine pruning and a concomitant loss of LTD are more prominent in iSPNs than dSPNs [15,16,17,20]. Although marked pathophysiological adaptations affect dSPNs too [14,19,25,26], spine loss is delayed and more variable in these neurons [27,30]. Based on these considerations, we chose to focus our study on the spine loss and synaptic dysfunction affecting the iSPNs.

Our results reveal that caspase-3 is promptly and transiently activated in the striatum upon DA denervation, and that iSPNs show loss of dendritic spines and LTD disturbance at the same early time points. Treatment with Q-VD-OPh, a third-generation broad-spectrum caspase inhibitor with therapeutic potential [31], prevented the loss of dendritic spines and LTD in iSPNs without interfering with the ongoing degeneration of DA neurons. Our study, therefore, points to a connection between spine loss and early synaptic dysfunction of iSPNs and non-apoptotic activation of caspase-3 in the parkinsonian striatum.

## 2. Results

### 2.1. Transient and Early Non-Apoptotic Caspase-3 Activation in the DA-Denervated Striatum

To investigate the time course of caspase-3 activation, we performed 6-OHDA injections in the medial forebrain bundle (MFB) and euthanized the mice at 3 to 28 days post-lesion. Striatal sections were immunostained using an antibody recognizing both pro- and cleaved caspase-3, as in [23]. On the side contralateral to the lesion (“intact”), the levels of caspase-3 immunostaining were low, and comparable to those found in sham-operated mice (Figure 1A–F, left). In contrast, a large number of cell bodies and processes were immunoreactive for caspase-3 on the DA-denervated side (“lesion”), particularly at early time points (Figure 1A–F, right). The peak of caspase-3 expression in the DA-denervated striatum was detected at five days post-lesion (1.75 +/− 0.19-fold over intact side, Figure 1C,G), after which the signal steadily declined. At five days post-lesion, the caspase-3 immunostaining co-localized with the astrocyte markers GFAP and S100β (Figure 1H,I) and the microglial marker IBA1 (Figure 1J). To assess whether caspase-3 also was expressed in iSPNs, we labeled this neuron population using BAC-*adora2a*-Cre mice and a Cre-inducible AAV-eGFP vector. Upon examining these mice five days after a 6-OHDA lesion, we found labeling for caspase-3 in multiple puncta along iSPN dendrites (Figure 1K,L).

Why is caspase-3 elevated so shortly after the 6-OHDA lesion? Due to its known role in axon degeneration, a likely reason could be the involvement of caspase-3 in the breakdown of DAergic fibers that richly innervate the striatum. To test this hypothesis, we performed Western blot analysis of striatal tyrosine-hydroxylase (TH) levels, a widely used dopaminergic cell marker. Our analysis shows a rapid, exponential decay of striatal TH levels after the 6-OHDA injection, with the bulk already being lost three days post-lesion (0.29 +/− 0.03 and 0.20 +/− 0.04 relative to the intact side, at three and five days post-lesion, respectively; Figure 2A). This is in line with the report by Rentsch and colleagues, showing that striatal TH vanishes quickly even though the degeneration of dopaminergic neurons in the substantia nigra takes weeks to become established [32]. We then examined the same striatal samples for caspase-3 activation, measured as the ratio of cleaved caspase-3 fragments (17 and 19 kDA bands) over the pro-caspase-3 protein (35 kDa band). This ratio was significantly increased at five days (1.86 +/− 0.32 relative to intact side) but not three days post-lesion (1.21 +/− 0.12 relative to intact side; Figure 3B), indicating that the degeneration of TH axons precedes the peak of caspase-3 activation.

When associated with apoptosis, caspase-3 activation shows a predominantly nuclear localization. However, in the DA-denervated striatum, caspase-3 immunostaining occurred mostly in filamentous structures. Accordingly, using terminal deoxynucleotidyl transferase-mediated biotinylated dUTP nick end labeling (TUNEL), we did not detect any apoptotic cells in DA-denervated striatal sections (Figure 2D,D’), whereas we found many intensely stained nuclei along the 6-OHDA injection track in the caudal diencephalon (Figure 2C,C’). The lack of striatal apoptosis is in keeping with previous studies finding a normal number of neurons in the MFB lesion model [22,33].

Taken together, these results indicate that the degeneration of dopaminergic fibers is promptly followed by prominent and transient non-apoptotic activation of caspase-3 in the striatum, with conspicuous upregulation of caspase-3 protein levels in both iSPNs and glial cells.

### 2.2. Caspase Activation Mediates Early Structural and Synaptic Changes in DA-Denervated iSPNs

Spine loss and synaptic deficits are prominent in hippocampal neurons in models of Alzheimer’s disease, and caspase-3 is an important mediator of these changes [6,13]. Loss of dendritic spines and an inability to induce LTD are also observed in iSPNs after DA-denervation [14,15,16,17,18,19,20]. However, previous studies have not established when exactly these changes first appear. We therefore investigated if the loss of iSPN dendritic spines and LTD deficits already occur during the time window of marked caspase-3 activation.

At 5–6 days post-6-OHDA lesion, dendritic spine density was significantly reduced in iSPNs (1.36 +/− 0.06 and 1.03 +/− 0.04 spines per µm in sham and 6-OHDA + vehicle, respectively; Figure 3A,B). To evaluate if the activation of caspase-3 is causally linked to this spine loss, we co-treated 6-OHDA-lesioned mice with an effective dose of the systemically active pan-caspase-inhibitor Q-VD-OPh (10 mg/kg, s.c., twice daily on days 1-5 post-lesion; [34,35]). Interestingly, in the treated mice, iSPN spine density was not different from that in sham-lesioned controls (1.28 +/− 0.02 spines per µm; Figure 3A,B). Blocking caspase activation therefore prevents the loss of iSPN spines induced by DA denervation.

In PD models, the structural reorganization of iSPNs is accompanied by a deficit in LTD formation [14,17,18,20,21]. To induce LTD, we used an established protocol based on pairing high-frequency stimulation (HFS) of cortical afferents with postsynaptic depolarization (Figure 3C). In sham-lesioned mice, this protocol reduced evoked EPSCs in iSPNs by >30% (Figure 3D–F, black). This was accompanied by an increase in the paired-pulse ratio (PPR; 1.32 +/− 0.09 and 1.45 +/− 0.10 PPR at baseline and post HFS, respectively; Figure 3G). The PPR was measured by delivering two stimuli at a short interval (50 ms) and then dividing the amplitude of the second EPSC by the amplitude of the first. PPRs are a well-described indicator of synaptic release probabilities [36] and an increase in the PPR (indicating a reduced probability of presynaptic glutamate release) is expected for this form of LTD [37,38,39]. Interestingly, both LTD formation and the concomitant PPR increase were lost already 5–6 days after DA-denervation (1.06 +/− 0.06 EPSC amplitude relative to baseline; 1.16 +/− 0.05 and 1.18 +/− 0.05 PPR at baseline and post HFS, respectively; Figure 3D–G, magenta). However, upon cotreatment with the caspase-inhibitor Q-VD-OPh, HFS-induced LTD and concomitant PPR change did not differ from those in sham-lesioned controls (0.58 +/− 0.04 EPSC amplitude relative to baseline; 1.01 +/− 0.02 and 1.16 +/− 0.05 PPR at baseline and post HFS, respectively Figure 3D–G, cyan).

To determine whether Q-VD-OPh treatment had affected the extent of dopaminergic degeneration, we measured striatal TH levels by optical density analysis and the number of TH-positive neurons in the substantia nigra pars compacta (SNc) by unbiased stereological cell counting. In line with a previous study, we find that the loss of striatal TH-positive fibers precedes the loss of TH-positive cells in the SNc [32]. Additionally, striatal TH levels were reduced by over 90% regardless of the treatment (99 +/− 1.7 %, 4 +/− 0.6% and 3 +/− 0.5% of the intact side in Sham, 6-OHDA + vehicle and 6-OHDA + QVD, respectively; Figure 4A,C). Stereological counts of TH-positive cells in the SNc revealed a similar extent of DA neuron loss in 6-OHDA lesioned mice treated with Q-VD-OPh or vehicle (99 +/− 8%, 67 +/− 8% and 65 +/− 7% of the intact side in Sham, 6-OHDA + vehicle and 6-OHDA + QVD, respectively; Figure 4B,D). These data show that Q-VD-OPh treatment had not interfered with the process of dopaminergic degeneration induced by 6-OHDA.

Taken together, these results indicated that spine loss and the inability to form HFS-LTD affect iSPNs early after DA-denervation, coinciding with a period of high caspase-3 activity in the striatum. Interestingly, both alterations can be prevented by pharmacological treatment with a caspase inhibitor that does not modify the extent nor the pattern of nigrostriatal dopaminergic denervation.

## 3. Discussion

Our results show that transient, non-apoptotic activation of caspase-3 in the DA-denervated striatum temporally coincides with dendritic spine pruning and synaptic plasticity deficits in iSPNs. Treatment with a pan-caspase inhibitor prevented the loss of iSPN spines and preserved HFS-LTD, although it did not protect against the 6-OHDA-induced dopaminergic degeneration.

Previous studies addressing the role of caspase-3 in PD models have focused on its possible involvement in the apoptotic death of DA-producing neurons [23,40,41,42,43,44,45,46,47]. Thus, caspase-3 activation has been shown to mediate the apoptosis of nigral dopaminergic neurons in MPTP models [23,44,45,47], although results from 6-OHDA lesion models have varied between studies [40,41,42,43,46]. Few studies have investigated the expression and/or activity of caspase-3 in the striatum. One study reported that caspase-3 is apoptotically activated in the striatum of MPTP-treated mice seven days after the last MPTP dose [23], and a second study using a partial 6-OHDA lesion model observed expression of active caspase-3 in enkephalin-positive striatal neurons at four weeks post-lesion [22]. This upregulation of caspase-3 was reported to occur in the absence of cell loss, and its functional significance remained unclear [22]. We provide here the first evidence that non-apoptotic caspase-3 activation mediates early synaptic adaptations in iSPNs deprived of their dopaminergic afferents.

It is well established that, in both PD patients and DA-denervated animals, SPNs undergo a process of dendritic regression and loss of spines [16,48,49,50,51,52]. Previous studies in 6-OHDA-lesioned mice observed that the spine loss primarily occurs in iSPNs [15,16], or in the rare sub-population of SPNs expressing both the D1 and D2 receptors [53]. Although dSPNs undergo many modifications, spine pruning occurs only late in this cell population [30], presumably depending on a different mechanism. However, what does a loss of iSPN dendritic spines signify? An influential hypothesis states that the pruning of spines serves as a homeostatic response to rebalance the activity of iSPNs, which become hyperactive in the absence of D2 receptor stimulation [54]. Yet, maladaptive implications cannot be excluded since iSPN spine pruning concurs with loss of corticostriatal synapses and deficits in synaptic plasticity.

The present study provides valuable new insights into the temporal evolution and molecular mechanisms of iSPN spine loss. Our data show that spine pruning and LTD disruption are present in iSPNs already five days post lesion. This is a surprisingly early time point if one considers that the degeneration of nigral dopaminergic neurons and other SPN adaptations take weeks to manifest completely. The loss of iSPN dendritic spines is therefore very rapid, most likely directly triggered by the loss of DA afferents and the ensuing lack of D2 receptor stimulation. In line with this interpretation, iSPN dendritic spine density has been reported to decrease after pharmacological DA depletion with reserpine [15] or treatment with the D2 receptor-antagonist haloperidol, even when administered for only 5 days [55].

In a seminal study, Day et al. showed that selective elimination of iSPN spines in PD models is due to increased Ca^2+^ influx via CaV1.3 channels, which are positioned in spines. Accordingly, SPNs of CaV1.3-knockout mice have higher spine densities and are resistant to 6-OHDA-induced spine loss [15]. However, the mechanisms linking intraspine Ca^2+^ levels to spine removal have not been established. Based on the present results, we propose that caspase-3 activation provides a critical link, in a fashion similar to what has been described for hippocampal neurons. In these neurons, dendritic Ca^2+^ influx leads to sequential activation of caspase-9 and caspase-3, followed by spine pruning [7,8]. Caspase-3 could initiate spine removal by cleaving cytoskeletal components, such as actin [56] and actin-regulating proteins [57]. It also increases the activity of calcineurin [58], a Ca^2+^-dependent phosphatase regulating cytoskeletal proteins [59]. Another potential role of caspase-3 may consist in promoting the release of chemoattractant factors necessary for phagocytic cells to engulf and remove damaged cellular components. This function of caspase-3 has been demonstrated in a model of dendritic pruning in *Drosophila* [60] and in various cellular models of apoptosis [61].

Besides the roles in neuronal plasticity, caspase-3 also activates microglia [11] and pushes astrocytes towards a reactive phenotype [9,10]. This glial activation can itself orchestrate the removal of dendritic spines [62]. We observed that many glia cells were labeled for caspase-3 in the DA-denervated striatum. Thus, a caspase-regulated glial contribution to the process of spine pruning seems likely. However, if spine loss was solely dependent on the lesion-induced glial activation, it would equally affect the two SPN populations, whereas it is well established that spine pruning selectively affects iSPNs for up to six weeks after 6-OHDA infusion, at least in the MFB lesion model [16,27]. Moreover, the observation that selective iSPN spine loss can be induced using pharmacological treatments that do not cause any significant glial activation [15,55] speaks in favor of an intraneuronal triggering mechanism.

The inability to induce HFS-LTD in iSPNs after DA denervation has been reported previously [14,17,18]. Here, we add that this deficit already occurs five days after the 6-OHDA lesion. In parallel with the preservation of dendritic spines, pharmacological caspase inhibition preserved the ability to induce LTD in iSPNs. HFS-LTD is mediated by postsynaptic endocannabinoid release and activation of presynaptic CB1 receptors (CB1Rs), which leads to an inhibition of transmitter release from the glutamatergic axon terminal. The CB1Rs are, however, only expressed on corticostriatal and not thalamostriatal terminals [39]. This is remarkable because we have previously shown that specifically corticostriatal synapses are pruned in iSPNs in this model of PD [16]. It is tempting to speculate that the selective loss of cortically innervated dendritic spines causes the disruption of this form of corticostriatal LTD by simply eliminating its structural substrate, and that Q-VD-OPh treatment preserves HFS-LTD in iSPNs by maintaining the integrity of their dendritic spines and the connected corticostriatal terminals. Interestingly, a previous ultrastructural study has revealed a marked loss of VGluT1-positive corticostriatal presynaptic terminals in MFB-lesioned rodents [63], and our previous study showed that denervation-induced iSPN spine pruning goes hand in hand with a loss of corticostriatal synapses [16]. Additionally, several studies show that input neurons from the motor cortex undergo various changes following DA-denervation [64,65,66]. The present set of data does not allow us to exclude that loss of iSPN spines and HFS-LTD are two independent consequences of DA denervation. The disadvantage of this viewpoint is, however, that it fails to readily explain why caspase-inhibition restores both phenomena, since our data show that DA innervation is not spared by such treatment.

We chose Q-VD-OPh because the more specific caspase-3 inhibitors like Z-DEVD-FMK do not cross the blood–brain barrier and would need to be injected directly into the brain. While being principally feasible, this approach is disadvantageous, as delivery of compounds directly into the striatum through cannulas is prone to induce tissue trauma and inflammatory responses that can confound the data. Q-VD-OPh on the other hand is a caspase inhibitor with therapeutic potential [67], and has been tested in several disease models. For example, it has been found to delay disease progression in AIDS models [68], reduce brain damage in models of stroke [69], and attenuate Alzheimer’s pathology [35]. In the MPTP model of PD, systemic treatment with Q-VD-OPh reduces DA cell death and spares TH positive fibers in the striatum when low doses of MPTP are used [70], potentially by preventing caspase-3-mediated apoptosis of nigral dopaminergic neurons [71]. In our 6-OHDA model, however, Q-VD-OPh did not prevent the loss of dopaminergic neurons or TH fibers in the striatum. In agreement with a previous study [41], our results therefore indicate that the 6-OHDA-induced dopaminergic degeneration is largely independent of caspase-3-mediated apoptosis.

The functions of caspase-3 beyond apoptosis have started to be investigated relatively recently. After the pioneering results obtained in glutamatergic pyramidal neurons [6,7,8,13] we here provide the first evidence that also in the GABAergic, principal neurons of the striatum, caspase-3 serves non-apoptotic roles in mediating the pruning of dendritic spines and the associated synaptic alterations in a model of PD. Our results further show that loss of spines and dysfunctional LTD go hand in hand in iSPNs after DA denervation, which is reminiscent of the reports from other pathologies, such as hippocampal neurons in Alzheimer’s disease models [6,13] and accumbal neurons in ethanol-dependent rats [72]. In future studies, it will be interesting to determine whether caspase-3 mediates the synaptic remodeling of SPNs also in non-pathological situations, such as during development or striatum-dependent learning tasks.

## 4. Materials and Methods

### 4.1. Animals

We used heterozygous BAC-*adora2a*-GFP and BAC-*adora2a*-Cre transgenic mice (GENSAT project) on a C57BL/6 background and their non-transgenic littermates (bred in the Animal Facility of Lund University, Biomedical Center). Mice were housed under a 12 h light/12 h dark cycle with free access to food and water. The animals were at least nine weeks old at the beginning of the experiments and a total of 80 mice of both sexes were used. Animal numbers and experimental layouts are provided in Appendix A. All experiments were approved by the Malmö-Lund Ethical Committee on Animal Research (ethical permit number M47-16, granted on 20 April 2016) and were conducted in adherence with the EU directive 2010/63/EU.

### 4.2. Stereotactic Surgeries

Unilateral 6-OHDA lesions and adeno-associated virus (AAV) injections were performed as described previously [73]. In brief, mice were anesthetized with a mixture of 4% isoflurane in air (Isoba vet, Apoteksbolaget) and placed in a stereotaxic frame (Kopf Instruments, Model 923-B mouse Gas Anethesia Head Holder, with 922 Ear Bars for mouse). 6-OHDA hydrochloride was dissolved in 0.02% ascorbic acid (3.2 μg free-base per μL) and 0.7 μL was injected at 0.2 µL/min in the median forebrain bundle at following coordinates: AP = −0.7, ML = −1.2, DV = −4.7, tooth bar: −4.0 [73]. For marking iSPNs, we injected AAV5-hSyn-DIO-EGFP-WPRE into the striatum at two injection sites (0.5 μL each): AP = +1, ML = −2.1, DV = −2.9; and AP = +0.3, ML = −2.3, DV = −3.0 [73]. After the surgery, the wound was closed with tissue glue and the animal received an s.c. injection of an analgesic (Marcaine, AstraZeneca; 2.5 mg/mL, 1 µL/g b.w.). To prevent dehydration, mice received s.c. injections of sterile glucose-ringer acetate (0.6 mL) immediately after the surgery. Further injections and dietary supplementations were given as necessary in the days post-lesion. Control mice received a sham-lesion surgery (capillary was inserted without delivering any injection). The success of the lesion was verified by TH-immunohistochemistry after the experiments.

### 4.3. Western Immunoblotting

Striatal samples were analyzed as described previously [74]. Animals were anesthetized with pentobarbital (500 mg/kg, i.p.) and briefly perfused with artificial cerebrospinal fluid (aCSF, see below). The brains were rapidly extracted and cut into thick sections on a Vibratome (Leica VT1200S) in ice-cold aCSF. The striatum was dissected out, transferred into tubes, rapidly frozen on crushed dry ice and stored at −80 °C until used. Samples were homogenized in RIPAlysis buffer, containing 65 mM Tris-base, 150 mM NaCl, 1% Triton-X, 0.25% sodium deoxycholate, 1 mM EDTA, and phosphatase and protease inhibitors (“phosSTOP” and “Complete, mini, EDTA-free,” Roche Applied Science, Penzberg, Germany). Protein concentration was determined using the BCA Protein Assay Kit (Pierce #23225, Thermo Scientific, Uppsala, Sweden). Samples were run on a 7.5% SDS gel and transferred on PVD membranes. Membranes were blocked with 5% nonfat dry milk and incubated overnight with the following antibodies: anti-tyrosine hydroxylase (Pel-Freez, P40101), anti-caspase-3 (Cell Signaling, #9662, Lot 17, Danvers, MA, USA) and beta-actin (Sigma, A-3854). After washing, the membranes were incubated with HRP-linked secondary antibodies (anti-biotin HRP-linked antibody Cell Signaling, #7075; anti-rabbit-IgG HRP-linked antibody, Cell Signaling, #7074) and bands were visualized by chemiluminescence using an ECL kit (Pierce, #32106, Thermo Scientific). Images were acquired using a CCD camera (LAS1000 system, Fuji Films, Tokyo, Japan) and analyzed using ImageJ (National Institutes of Health, Bethesda, MD, USA). After image acquisition, membranes were stripped and re-blotted for loading controls.

### 4.4. Immunohistochemistry and Optical Density Analysis

Mice were anesthetized with sodium pentobarbital (240 mg/kg, i.p., Apoteksbolaget, Uppsala, Sweden) and transcardially perfused with 0.1 M phosphate-buffered saline (PBS, pH 7.4), followed by ice-cold 4% paraformaldehyde in PBS. Brains were post-fixed in the same solution overnight. Coronal sections of 30 μm thickness were cut while immersed in PBS at 4 °C using a vibratome (VT 1200S, Leica, Wetzlar, Germany). Sections were stored at −20 °C in a non-freezing solution (30% ethylene glycol and 30% glycerol in 0.1 M phosphate buffer).

Bright-field immunohistochemistry for TH and caspase-3 was performed according to previously described protocols [75], with minor modifications. Caspase-3 expression was studied using a primary antibody that recognizes both the pro- and cleaved caspase-3 (mouse anti-caspase-3, BD Transduction Laboratories, cat # BD 611049; 1:100). The immunostaining specificity was confirmed through omission of either primary or secondary antibodies, and through comparisons with the regional and cellular expression patterns produced by other caspase-3 antibodies (BD Transduction Laboratories #BD610322; Cell Signaling Technology, #9662; SCBT, #56053). Briefly, sections were rinsed in 0.02 M potassium phosphate-buffered saline (KPBS, pH 7.4) and transferred to a citrate buffer (pH 6.0) for antigen retrieval (carried out at 80 °C for 30 min). After three rinses with KPBS, sections were pretreated for 5 min in a solution of 3% hydrogen peroxide (H_2_O_2_) and 10% methanol to quench endogenous peroxidase activity. Sections were then pre-incubated for 1 h in a blocking buffer (5% normal horse serum in KPBS with 0.1% Triton-X, KPBS-T). This was followed by overnight incubation at 4 °C with the primary antibody. Next, sections were rinsed and incubated with biotinylated horse-anti-mouse secondary antibodies (Vector Laboratories, # BA2001, 1:200, Burlingame, CA, USA) for two hours, followed by incubation in an avidin-biotin-peroxidase solution (Vectastain Elite ABC, Vector Laboratories) for one hour at room temperature. The immunocomplexes were visualized using 3,3-diaminobenzidine (DAB) and H_2_O_2_ (0.05% and 0.04%, respectively). Finally, sections were rinsed in KPBS-T, mounted onto chromalum-coated slides, and coverslipped using DPX mounting medium. Images were taken on a Nikon Eclipse 80i microscope under a 20× objective (0.50 NA). Sections immunostained for TH (Pel-Freez, P40101-150; 1:1000) were digitized at low magnification (4× objective, NA 0.10) per our established methods [75,76]. Densitometric measurements of caspase-3 and TH immunostaining were carried out on predetermined regions of interest using ImageJ. Results from the side ipsilateral to the lesion were expressed as a percentage of the values measured on the contralateral side in each section.

Sections spanning the striatum were also processed for dual-antigen immunofluorescence to detect caspase-3 in iSPNs, microglia and astrocytes using the following primary antibodies: rabbit anti-caspase-3 (Cell Signaling Technology, #9662, 1:200) or mouse anti-caspase-3 (BD Transduction Laboratories, BD611049, 1:50), mouse anti-S100β (Sigma Aldrich, S2532, 1:100, St. Louis, MO, USA) and chicken anti-Iba1 (Synaptic Systems, #234009, 1:500, Göttingen, Germany). Sections were rinsed in Trizma buffered-saline (TBS) prior to a one hour pre-incubation in a blocking solution consisting of 5% normal serum (from goat or donkey) in TBS with 0.1% TritonX (TBS-T). Primary antibodies were diluted in the same blocking solution and sections were incubated overnight at 4 °C. On the second day, sections were incubated in secondary antibody solution for 2 h at room temperature. We used the following antibodies: Alexa Fluor 488-conjugated goat-anti-rabbit (Invitrogen, 1:200, Waltham, MA, USA); Alexa Fluor 647-conjugated donkey-anti-mouse (Invitrogen, 1:200) and Alexa Fluor 647-conjugated donkey anti-chicken (Jackson Immuno Research, 1:200, West Grove, PA, USA). Secondary antibodies were diluted in the same blocking solution as the primary antibody. Alexa FluorTM 488 Tyramide SuperBoostTM Kit (Invitrogen, B40922) was used to improve the signal-to-noise ratio for primary antibody rabbit anti-caspase-3. After secondary antibody incubation, sections were rinsed in TBS, mounted and coverslipped with polyvinyl alcohol mounting medium with DABCO (PVA-DABCO, Sigma-Aldrich), and imaged using a confocal laser scanning microscope (LSM710 NLO, Zeiss, Jena, Germany).

### 4.5. Stereological Counts of Nigral DA Neurons

The number of TH positive cells was determined by unbiased stereology according to the optical fractionator method [77] using the protocol detailed previously [75,76]. In brief, four serial sections spanning the rostrocaudal extent of the substantia nigra pars compacta (from −2.9 to −3.6 mm posterior to bregma) were sampled in each animal. Analysis was performed using a Nikon 80i microscope with an x-y motorized stage controlled by NewCAST software (Visiopharm). The area of interest was outlined under 4× objective and counting of neurons was performed under 100× objective. The area sampling fraction and slice sampling fraction were set to 30% and 33.3%, respectively. The total number of TH-positive neurons in the substantia nigra pars compacta was estimated in each hemisphere using the optical fractionator formula, i.e., number of neurons = 1/ssf (slice sampling fraction) × 1/asf (area sampling fraction) × 1/tsf (thickness sampling fraction) × Σ(number of objects counted) [77].

### 4.6. Visualization of Apoptosis (TUNEL)

Terminal deoxynucleotidyl transferase-mediated biotinylated dUTP nick end labeling (TUNEL) was carried out in paraffin-embedded sections of 5 µm thickness, prepared from 3 mice that had been euthanized at 5 days post-lesion. The procedure was applied to both, striatal and midbrain sections encompassing the medial forebrain bundle (target of 6-OHDA infusion). We used a fluorescence-based TUNEL in situ cell death detection kit (FITC; Roche) according to the manufacturer’s instructions and previous studies [24,78]. In brief, after preheating at 60 °C for 30 min, sections were rehydrated through a series of washing in xylene, absolute ethanol, graded ethanol and deionized distilled water. Following section rehydration, antigen retrieval in Proteinase K (20 µg/mL in 10 mM Tris/HCl, pH 7.4) was performed for 15 min at room temperature. Thereafter, the enzyme solution and label solution from the TUNEL kit were applied for 60 min at 37 °C. After washing three times with PBS the sections were counterstained with TO-PRO-3 Iodide 642/661 (Invitrogen, T3605) at a final concentration of 1 µM for 5 min at room temperature. Following a last washing step, sections were coverslipped with PVA-DABCO. Not all TUNEL-positive cells co-labeled with TO-PRO-3, which likely indicates different stages of apoptosis, as it has been shown that TO-PRO-3 labels early apoptotic and necrotic cells differentially [79].

### 4.7. In Vivo Caspase Inhibition with Q-VD-OPh

Q-VD-OPh (APExBio, Cat.No.: A1901) was dissolved first in DMSO and then diluted to the final concentration (2 mg/mL, 50% DMSO) with saline. Animals received injections of 10 mg/kg s.c., twice daily, with an injection volume of 5 mL/kg body weight.

### 4.8. Electrophysiology and Two-Photon Imaging

Acute brain slices were prepared as described previously [74,80]. In brief: mice were deeply anesthetized with Pentobarbital (65 mg/kg, i.p.) and shortly perfused with ice-cold aCSF containing (in mM): 124.0 NaCl, 3.0 KCl, 2.0 CaCl_2_, 1.0 MgCl_2_, 26.0 NaCO_3_, 1.0 NaH_2_PO_4_ and 16.66 glucose. The osmolarity was 300–310 mOsm/L and the pH 7.4. Oxygenation and pH were maintained by gassing the aCSF with 5%/95% CO_2_/O_2_. Parasagittal slices (275 μm) were cut on a vibratome (VT1200s, Leica, Germany) and incubated at 34 °C for 30 min. Afterwards, the temperature was allowed to return to room temperature. LTD measurements were done at 30 °C.

The iSPNs in the dorsolateral striatum were identified by somatic eGFP expression. Patch pipettes (3–5 MOhm) were pulled from thick-walled borosilicate glass on a Sutter P-97 puller. Recordings were sampled at 10–20 kHz using a Multiclamp 700B amplifier (Molecular Devices, San Jose, CA, USA) and digitized (Digidata 1440, Molecular Devices, USA). Data were analyzed in pClamp (v.10, Molecular Devices, USA). For voltage clamp recordings the internal solution contained (in mM): 120 CsMeSO_3_, 5 NaCl, 10 TEA-Cl, 10 HEPES, 5 QX-314, 4 ATP-Mg and 0.3 GTP-Na. A stimulation electrode was placed in the dorsal striatum at the border with the corpus callosum. EPSCs were evoked at 0.05 Hz and cells were held at a potential of −70 mV in the presence of picrotoxin (50 µM). For LTD induction, synaptic inputs were stimulated with four trains (each 1 s at 100 Hz), spaced 10 s apart. During the stimulation, the cell was depolarized to 0 mV. All EPSCs were normalized to the average EPSC size of the respective baseline. Baseline EPSCS were not significantly different for the different groups: −241 +/− 15.6 pA, −269 +/− 25.4 pA and −386 +/− 77.9 pA, for Sham, 6-OHDA + vehicle and 6-OHDA + QVD, respectively (One-way ANOVA, F(2, 17) = 2.728, *p* = 0.0938; Bonferroni’s multiple comparison test). For PPR measurements, two electric stimuli were given at an interstimulus interval (ISI) of 50 ms. To calculate the PPR, the amplitude of the second EPSC was divided by the amplitude of the first EPSC. Cortico-striatal synapses typically paired-pulse facilitate with short ISI, hence show a PPR > 1, and changes in PPRs are commonly interpreted as changes in presynaptic release probabilities [36,37,38]. Access resistance was continuously monitored and recordings were discarded if the access resistance at the end of the experiment was altered by >20%.

Two-photon imaging was performed on a Zeiss 710 NLO with a MaiThai laser (Spectra Physics, Sweden), as previously described [48,80]. To reveal SPN morphology, 50 μM of AlexaFluor-568 was added to the internal recording solution and excited at 780 nm through a 63× water-immersion objective (1.0 NA, Zeiss). For spine analysis, dendritic regions of interest were imaged at mid-distance from the soma (typically 50–100 μm), avoiding the aspiny proximal and the very distal regions (>100 μm). Dendritic stretches of 25–35 μm length were optically sectioned in a z-stack, with an x-y pixel size of 0.053 × 0.053 μm^2^ and z-sections were spaced 0.65 μm apart. Typically, 2–3 dendrites were analyzed per cell and dendritic spines were manually counted using ImageJ, in a blinded manner, as described previously [80].

### 4.9. Statistical Analysis

Statistical analysis was carried out using GraphPad Prism 8 and tests are specified in the respective figure legends and detailed results are provided in Appendix A. A probability of *p* < 0.05 was considered statistically significant.

## 5. Concluding Remarks and Perspectives

This study presents three major findings: (1) the loss of dendritic spines and HFS-LTD in iSPNs occurs soon after striatal DA denervation, (2) caspase-3 is activated early and non-apoptotically in the same time frame, and (3) pharmacological caspase inhibition using a clinically relevant compound prevents the early synaptic alterations of iSPNs in a manner that does not require sparing or restoration of striatal DA fibers. These findings contribute new knowledge to the growing field of non-apoptotic caspase functions and an improved understanding of the striatal response to DA denervation. From a therapeutic perspective, our results point to a possible application of caspase inhibitors to prevent SPN alterations that are associated with the motor dysfunction of PD (reviewed in [27]). This suggestion, however, needs to be substantiated by new dedicated investigations.

## Figures and Tables

**Figure 1 ijms-23-05470-f001:**
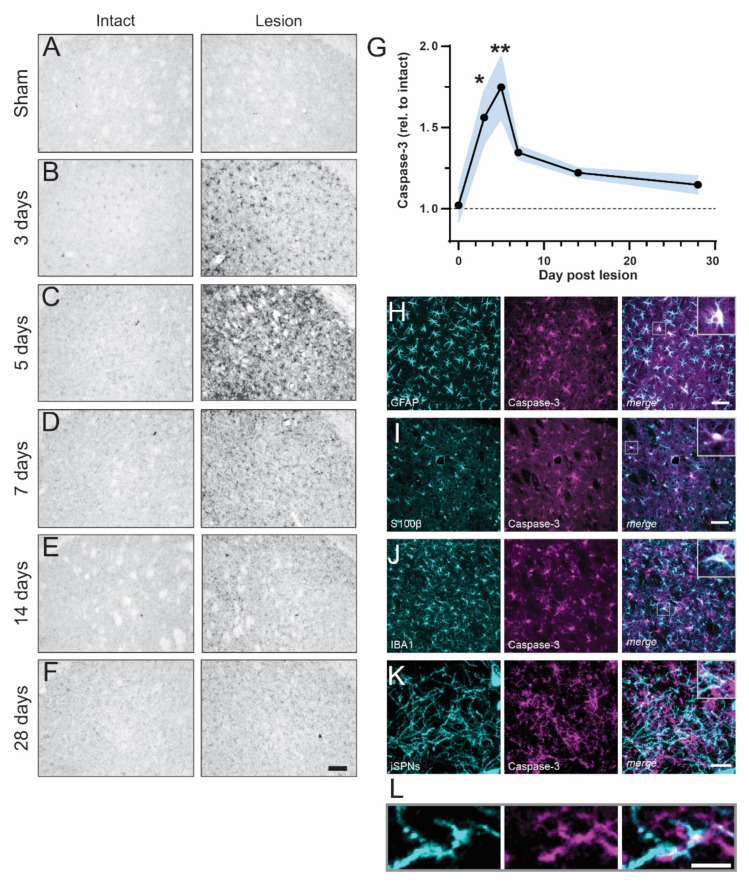
Caspase-3 is transiently upregulated in the DA-denervated striatum. (**A**–**F**) Striatal sections immunostained with a caspase-3 antibody depict the time course of caspase-3 activation in the DA-denervated striatum (right) relative to the control side (left) post 6-OHDA lesion. Scale bar: 100 μm. (**G**) Densitometric analysis reveals an increase of caspase-3 levels in the DA-denervated striatum at three- and five-days post-lesion as compared to sham-lesioned animals (represented by “Day 0” datapoint). * *p* < 0.05, ** *p* < 0.01 vs. Sham, ANOVA and post hoc Bonferroni test, N = 4–7. (**H**–**K**) Dual-antigen immunostaining demonstrates caspase-3 co-labeling with markers for astrocytes (GFAP and S100β; **H**,**I**), microglia (IBA1; **J**) and iSPN dendrites (eGFP; **K**). The latter were labeled using an AAV-based strategy (see methods). (**L**) Zoom-in of inset in (**K**) shows puncta of caspase-3 expression within iSPN dendrites. Scale bar: 20 µm (**H**–**J**), 10 μm (**K**), 3 µm (**L**).

**Figure 2 ijms-23-05470-f002:**
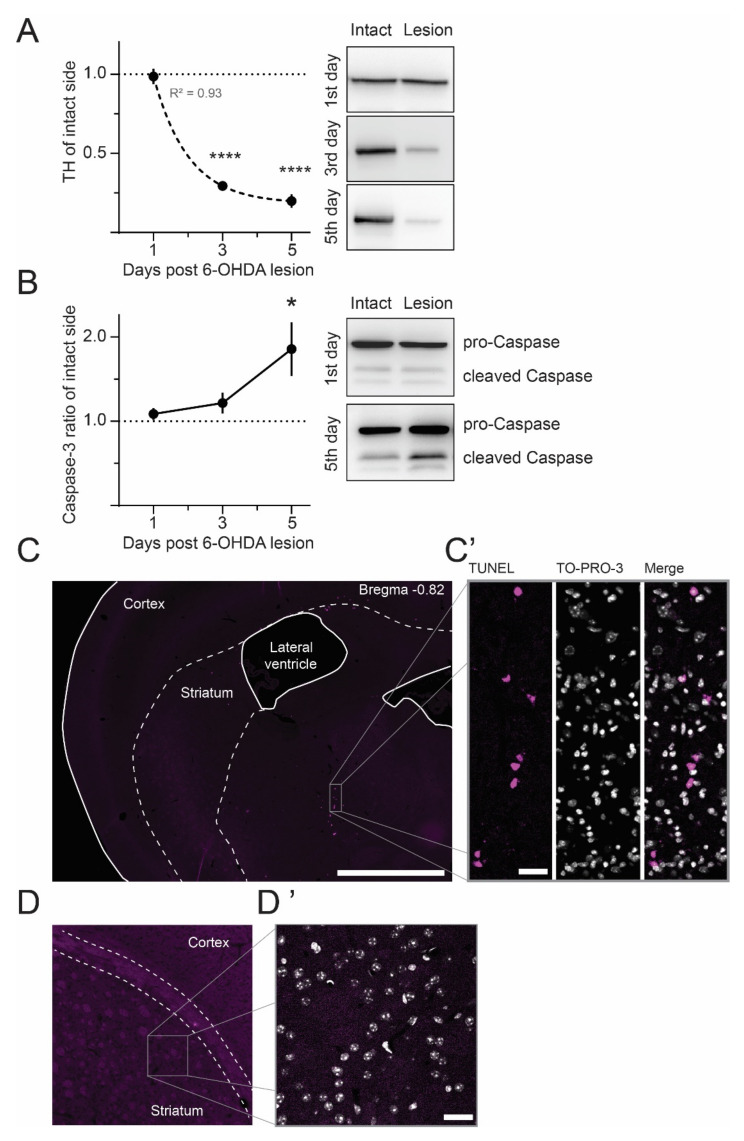
The peak of caspase-3 activation follows the loss of striatal DAergic innervation. (**A**) Western blot analysis of striatal TH levels shows an exponential decline of DAergic innervation between one and five days after the 6-OHDA lesion (dotted line depicts a one-phase decay fit with the given R² value). **** *p* < 0.0001 vs. one day post-lesion, ANOVA and post hoc Bonferroni test, N = 6–7. Right: Representative immunoblots with TH bands (60 kDa). (**B**) Caspase-3 activation was assessed as a ratio of the cleaved (19 and 17 kDa bands) over the uncleaved protein (35 kDa). The ratios are shown relative to the corresponding intact striatum and confirm a peak of caspase-3 activation at five days post-lesion. * *p* < 0.05 vs. one day post-lesion, ANOVA and post hoc Bonferroni test, N = 6–7. Right: Example of caspase-3-stained blots. (**C**,**D**) TUNEL staining was used to reveal apoptotic nuclei in sections through the caudal diencephalon (encompassing the MFB, **C**) and DA-denervated striatum (**D**). At the peak of caspase activation (five days post-6-OHDA lesion), no TUNEL positivity was found in the striatum (**D**,**D’**), whereas numerous cells displayed apoptotic features along the track of the 6-OHDA infusion (**C**,**C’**). (**C’**,**D’**) High magnification of (**C**,**D**), as indicated. TUNEL (magenta) is overlaid with TO-PRO-3 counterstain of neuronal nuclei (gray). Among the TUNEL-positive cells, some are colabeled with TO-PRO-3 and others are not, indicating different stages of apoptosis. Scale bars: 1 mm (**C**), 40 µm (**C’**) 10 µm (**D**’).

**Figure 3 ijms-23-05470-f003:**
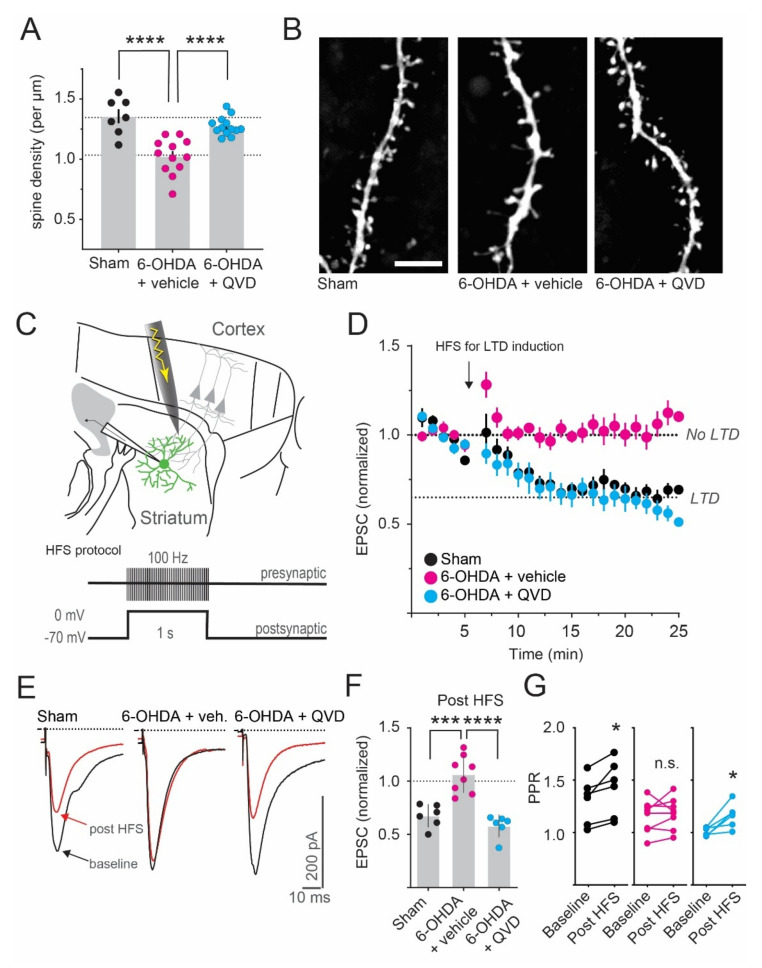
Early synaptic deficits after DA denervation are prevented by pharmacological caspase inhibition. (**A**) Dendritic spine density of iSPNs is reduced 5–6 days post-6-OHDA lesion. This loss is prevented by systemic treatment with the caspase inhibitor Q-VD-OPh (“QVD”). **** *p* < 0.0001 vs. 6-OHDA + vehicle, ANOVA and post hoc Bonferroni tests. N = 7–13 cells. (**B**) Two-photon images of iSPN dendrites, visualized after dye-filling the neurons through the patch pipette. Maximum-intensity projections of dendrites from controls and 6-OHDA-lesioned mice treated with vehicle or QVD are shown from left to right. Scale bar: 5 μm. (**C**) Sketch of the LTD recording paradigm. GFP-positive iSPNs were patched in the dorsolateral striatum and the stimulus electrode was placed near the border of the cortex. To induce LTD, high-frequent input stimulation (1 s at 100 Hz) was paired with postsynaptic depolarization. (**D**) Corticostriatal EPSCs are depressed using this protocol in control iSPNs (black). However, EPSC amplitudes are unchanged after HFS in 6-OHDA lesioned mice at 5–6 days after lesion (magenta). Co-treatment with QVD rescues this deficit, and LTD is readily induced (cyan). (**E**) Example traces from the recordings. Single EPSCs are shown before (black) and after (red) HFS. (**F**) Quantification shows loss and rescue of HFS-LTD. *** *p* < 0.001, **** *p* < 0.0001 vs. 6-OHDA + vehicle, ANOVA and post hoc Bonferroni test, N = 6-8 cells (**G**) After HFS-LTD induction, paired-pulse ratios (PPRs) are increased in control iSPNs (black) and unchanged in the DA-denervated striatum (magenta). Treatment with QVD rescues the LTD induction, with concomitant increase in PPRs (cyan). * *p* < 0.05, paired *t*-test, N = 6-8 cells.

**Figure 4 ijms-23-05470-f004:**
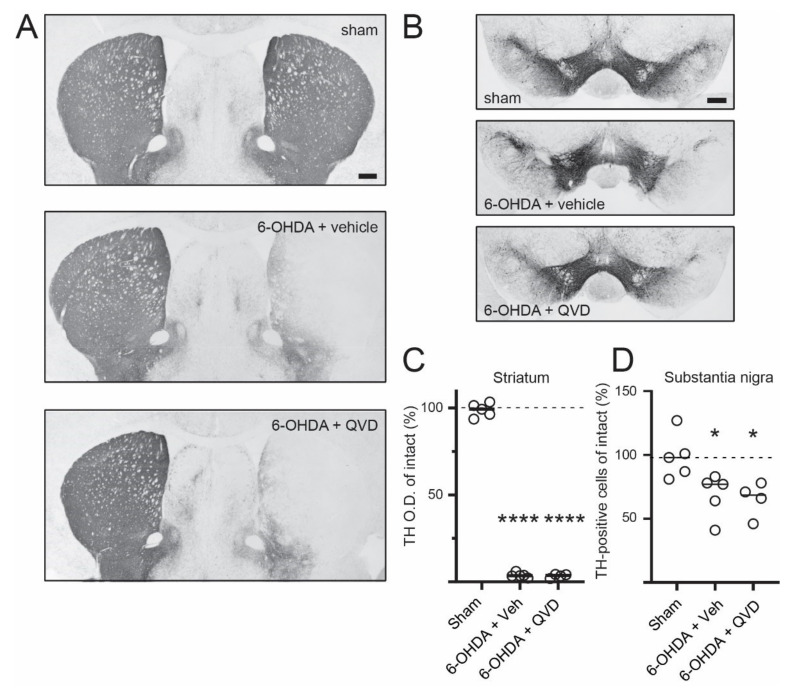
Q-VD-OPh treatment does not protect against 6-OHDA-induced dopaminergic degeneration. (**A**,**B**) Low-magnification images of striatal (**A**) and nigral (**B**) sections immunostained for TH, with the side ipsilateral to the 6-OHDA lesion on the right. Scale bar: 400 μm (**C**) Densitometric analysis shows that the loss of striatal DAergic innervation does not differ between vehicle- and QVD-treated animals. **** *p* < 0.0001 vs. sham, ANOVA and post hoc Bonferroni test, N = 4–5. (**D**) Stereological cell counts show that the 6-OHDA-induced loss of nigral DA neurons is not prevented by QVD treatment. * *p* < 0.05 vs. sham, ANOVA and post hoc Bonferroni test, N = 4–5.

## Data Availability

The presented data are available upon reasonable request from the corresponding authors.

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
