# Peer review of "Non-Apoptotic Caspase-3 Activation Mediates Early Synaptic Dysfunction of Indirect Pathway Neurons in the Parkinsonian Striatum"

_ijms, 2022, doi:10.3390/ijms23105470_

Round 1
Reviewer 1 Report
The work presented by Fieblinger and colleagues is a very interesting study demonstrating the involvement of caspase-3 in spine loss and impairment of LTD in a PD mouse model. The manuscript is well written and the study seems to be conducted thoroughly. I have no doubts about the general conclusions.
Below I have listed some minor concerns.
Line 83:
"Upon examining these mice five days after a 6-OHDA lesion, we found labeling for caspase-3 in multiple iSPN dendrites (Figure 1K)."
I cannot see colocalization of caspase-3 with iSPN dendrites in Fig. 1K. For my eyes they seem completely separated, only crossing each other (the white areas).
Fig. 1G: You reported the outcome of the post-hoc Bonferroni test, but not the ANOVA main effect. This can be found in the "ANOVA results" page in Prism under "ANOVA table". Can you please give the F-values, degrees of freedom and corresponding p-value, as it is provided in Prism, for every factor used. For example: F(1, 24)=12.6; p<0.0001 for factor XX.
Can you please explain the blue and grey shades of the curves? The line at 1.0 with the grey shade seems to be the sham line (please label the lines), but there are no data points attached. This means that you had only one sham value and used a one-way ANOVA to compare all lesion timepoints to the one sham timepoint - is this correct?
Fig. 2A+B: Same as for Fig. 1G: Please give ANOVA main effect values.
Fig. 2D: I don't understand where D is located in the brain. The figure legend says it's in the midbrain, encompassing the MFB. But why is the internal capsule mentioned, which is in the forebrain? It would be helpful to show a schematic overview of the whole section and point out where exactly the detail is.
Fig. 2D': I don't understand the meaning of the blue arrows. Do they point to a double labeling of TUNEL and Topro-3? What does single labeling of TUNEL (without Topro-3) mean, then? The blue arrows are very small. When D' is magnified, the blue arrows can be seen clearly, but some of them seem not to be in the correct position, e.g. the third one from the left.
Fig. 3A: Please give ANOVA main effect values.
"****p<0.0001 vs. sham" should read "... vs. 6-OHDA+vehicle".
Can you please use the same font in all subpanels? In A+B+C the letters look compressed.
Please use either n= or N=, but not a mixture.
Fig. 3C: Figure legend: "...the stimulus electrode was placed near the border of the cortex." Methods (line 493): "A stimulation electrode was placed in the dorsal striatum, at the border with the corpus callosum." In the image, the stimulation electrode is drawn in the corpus callosum. Please doublecheck where the stimulation electrode really was and eliminate the contradictions.
Fig. 3D: For the sham and 6-OHDA+QVD groups, there is a strong downward trend already visible during the baseline recordings. The sham 5 min-EPSC is halfway down to LTD level even before HFS. Usually in LTD figures there is a very stable baseline and a sudden drop of EPSC amplitude after HFS. Here it looks as if HFS briefly interrupted the ongoing decrease of the EPSC amplitude. Can you please comment on this?
Fig. 3E, legend: "Single EPSCs are shown before (black) and after (red) HFS stimulation." Please delete the word "stimulation" as the S in HFS already stands for stimulation.
Fig. 3F: Please give ANOVA main effect values.
X-axis labels are missing.
Fig. 4B: The figure label should not obscure the area you want to show. The words "vehicle" and "QVD" cover the SNc.
Fig. 4C+D: Please give ANOVA main effect values.
Line 187-226: Figure 4 must read Figure 3.
Line 231-233: Figure 5 must read Figure 4.
Line 220:
"This was accompanied by an increase of the paired-pulse ratio (PPR; Figure 4G), indicating reduced probability of presynaptic glutamate release, as expected for this form of LTD."
For readers not familiar with electrophysiology it would be helpful to describe the PPR paradigm in the methods (e.g. time between the first and second stimulus, and which way the ratio is calculated). I looked it up and found it is 2nd amplitude/1st amplitude. If PPR is an indicator of glutamate release, a PPR >1 means that more glutamate is released after the second stimulus than after the first one - correct? A PPR increase should therefore indicate an even greater glutamate release after the second stimulus, or a decrease of glutamate release after the first stimulus. From your interpretation I assume it is the latter.
To further complicate things: I found a publication (Manita et al., 2007 doi: 10.1016/j.brainres.2007.03.089), claiming that PPR in the hippocampus is not related to glutamate release. Can you explain a bit more about PPR in the text?
Line 230:
"Striatal TH levels were reduced by over 90% regardless of the treatment (Figure 5 A,C). Stereological counts of TH-positive cells in the substantia nigra revealed a similar extent of DA neuron loss in 6-OHDA lesioned mice treated with Q-VD-OPh or vehicle (Figure 5 B, D)."
According to Fig. 4D the loss of SNc neurons is more like 25%. This is odd, because one would not expect a reduction of the dopaminergic terminals of 90% with only a 25 % loss of cell bodies.
Line 321:
"It is tempting to speculate that the selective loss of cortically-innervated dendritic spines causes the disruption of this form of corticostriatal LTD by simply eliminating its structural substrate, and that Q-VD-OPh treatment preserves HFS-LTD in iSPNs by maintaining the integrity of their dendritic spines."
It is indeed tempting to think along those lines. But wouldn't the EPSC amplitude be greatly reduced as well when corticostriatal synapses are lost? In Fig. 3E the EPSC seems to be preserved in the 6-OHDA model, but this is only one recording. In Fig. 3D+F the EPSC is normalized (to baseline? You should mention this in the methods.). Maybe it would be helpful to display the absolute EPSC amplitudes to substantiate the above claim.
What do you think, how many synapses participate in an EPSC as shown in Fig. 3E? If you remove, say, 50% of them, the EPSC will be 50% smaller. But the ability of plasticity of the remaining synapses will be intact, so in a normalized EPSC the amount of LTD should be the same. Or do you think that the remaining synapses would somehow stop doing LTD so that their combined influence on the postsynaptic neuron would not be diminished any further?
Another thing is that (at least for me) it is not clear if loss of a dendritic spine means that the attached presynaptic terminal is also lost. The alternative would be that a spine synapse transforms into a shaft synapse. The EPSC may not change much, but synaptic plasticity would be reduced, because for many plastic changes the spine head compartment is needed. Maybe this is an alternative explanation.
Anyway, I suggest that you add some more about the interesting relationship between spine loss and reduced LTD in the discussion, taking into account the presence or absence of presynaptic terminals.
Methods
(Line 352 ff.) 4.1 Animals
How many mice were used in total for this study? And can you give an overview about the sample sizes for the different experiments? There is some information in the figure legends, but in some (e.g. Fig. 3) the number of cells is given rather than the number of animals.
For readers not familiar with the methods used in this work, changing of some section headings would be helpful:
(Line 459) 4.6 "Visualization of apoptosis" instead of "Terminal deoxynucleotidyl transferase-mediated biotinylated dUTP nick end labeling"
(Line 474) 4.7 "Caspase inhibition" instead of "Q-VD-OPh treatment"
Or something along those lines.
Reviewer 2 Report
The manuscript presented by Tim Fieblinger et al., is interesting, however there are same critical points to explore:
The authors showed that caspase-3 activation concurs with a rapid loss of dendritic spines and deficits in synaptic long-term depression (LTD) in striatal projection neurons forming the indirect pathway. Interestingly, systemic treatment with a caspase inhibitor prevents both the spine pruning and the deficit of indirect pathway LTD without interfering with the ongoing dopaminergic degeneration.
- In this context the authors should investigate if this treatments affected the expression of crucial factors, such as: NGF, BDNF, FGF that are involved in plasticity.
- Several previous studies showed that these can be modulate by caspase inhibitors
At the same time the authors should extend the introduction including the role of these mentioned factors.
Suggested literature:
- Kim DH, Zhao X. BDNF protects neurons following injury by modulation of caspase activity. Neurocrit Care. 2005;3(1):71-6. doi: 10.1385/NCC:3:1:071. PMID: 16159102.
- Oshitari T, Adachi-Usami E. The effect of caspase inhibitors and neurotrophic factors on damaged retinal ganglion cells. Neuroreport. 2003 Feb 10;14(2):289-92. doi: 10.1097/00001756-200302100-00027. PMID: 12598748.
- Guo, J., Ji, Y., Ding, Y. et al. BDNF pro-peptide regulates dendritic spines via caspase-3. Cell Death Dis 7, e2264 (2016). https://doi.org/10.1038/cddis.2016.166
- Yu Hasegawa, Cao Cheng, Kenyu Hayashi, Yushin Takemoto, Shokei Kim-Mitsuyama, Anti-apoptotic effects of BDNF-TrkB signaling in the treatment of hemorrhagic stroke, Brain Hemorrhages,
Volume 1, Issue 2,2020,Pages 124-132,ISSN 2589-238X, https://doi.org/10.1016/j.hest.2020.04.003. - Mnich, K., Carleton, L. A., Kavanagh, E. T., Doyle, K. M., Samali, A., & Gorman, A. M. (2014). Nerve growth factor-mediated inhibition of apoptosis post-caspase activation is due to removal of active caspase-3 in a lysosome-dependent manner. Cell death & disease, 5(5), e1202. https://doi.org/10.1038/cddis.2014.173
- Cheng Y, Deshmukh M, D'Costa A, Demaro JA, Gidday JM, Shah A, Sun Y, Jacquin MF, Johnson EM, Holtzman DM. Caspase inhibitor affords neuroprotection with delayed administration in a rat model of neonatal hypoxic-ischemic brain injury. J Clin Invest. 1998 May 1;101(9):1992-9. doi: 10.1172/JCI2169. PMID: 9576764; PMCID: PMC508786.
- Cacialli P. Neurotrophins Time Point Intervention after Traumatic Brain Injury: From Zebrafish to Human. International Journal of Molecular Sciences. 2021; 22(4):1585. https://doi.org/10.3390/ijms22041585
Reviewer 3 Report
In this manuscript, the authors used a mouse model of clinically manifest Parkinson’s disease, and provided first evidence that caspase-3 is transiently activated in the striatum shortly after the degeneration of nigrostriatal dopaminergic projections.
This manuscript is interesting; unfortunately, this manuscript needs substantial improvements and corrections before publishing may be possible.
General points:
Please add a list of abbreviations before References section to your manuscript.
Special points:
For better understanding of your research for readers, please add a time-line of all your experiments as a Figure 1 to your manuscript.
Keywords: please add also to keywords: mice.
Introduction
This manuscript should be substantially improved, i. e., by substantial references in the field:
Lines 24-25: please add multiple references at the end of this sentence.
Lines 36-38: please add multiple references at the end of this sentence.
Lines 52-53: please add multiple references at the end of this sentence.
Materials and Methods
4.1 Animals
Please describe all experimental groups and the number of the animals in each experimental group exactly as n = xxx.
Lines 353-355: please add a total amount of the animals used in all your experiments exactly.
Lines 357-358: please add also the date of the permission of all your experiments.
4.2 Stereotactic surgeries
Lines 361-362: please write out: AAV.
Lines 362-366: please say, which stereotaxic frame exactly did you use. Please also add a appropriate references for the stereotaxic coordinates used by you.
4.3 Western immunoblotting
Lines 377-378: please write out: aCSF.
4.5 Stereological counts of nigral DA neurons
Please describe this section more exactly.
4.6 Terminal deoxynucleotidyl transferase-mediated biotinylated dUTP nick end labeling
Please say, according to which group did you perform this method.
4.8 Statistical analysis
Please describe this section more exactly.
Results
Please add to your Results section also the appropriate values and not only the description of your results.
Discussion
Lines 256-257: please add multiple references at the end of this sentence.
Lines 261-264: please describe all these studies exactly.
Please add to your manuscript a “Conclusion” section making clear the importance of all your results for science and clinical practice.
Please add to your manuscript a “Future perspectives” section.
Round 2
Reviewer 2 Report
The authors have not improved the manuscript. For my opinion this article is not original at this time, I don't feel to support for publication.
Reviewer 3 Report
Dear authors,
thank you for your corrections.
As a last point, please add a detailed description of the Figure S1 in a Legend Figure S1.